# Genome Sequence and Evaluation of Safety and Probiotic Potential of *Lactiplantibacillus plantarum* LPJZ-658

**DOI:** 10.3390/microorganisms11061620

**Published:** 2023-06-20

**Authors:** Liquan Deng, Liming Liu, Tongyu Fu, Chunhua Li, Ningyi Jin, Heping Zhang, Chang Li, Yawen Liu, Cuiqing Zhao

**Affiliations:** 1School of Public Health, Jilin University, Changchun 130021, China; dengliquan@126.com (L.D.);; 2College of Animal Science and Technology, Jilin Agricultural Science and Technology University, Jilin 132101, China; aliuliming1984@126.com (L.L.); 15526615156@163.com (T.F.); chunhuali2009@sina.com (C.L.); 3Research Unit of Key Technologies for Prevention and Control of Virus Zoonoses, Chinese Academy of Medical Sciences, Changchun Veterinary Research Institute, Chinese Academy of Agricultural Sciences, Changchun 130122, Chinalichang78@163.com (C.L.); 4Department of Food Science and Engineering, Inner Mongolia Agricultural University, Huhhot 010010, China

**Keywords:** probiotic, *Lactiplantibacillus plantarum* LPJZ-658, genome sequencing, safety, probiotic properties

## Abstract

This study aims to systematically evaluate the safety of a novel *L. plantarum* LPJZ-658 explored on whole-genome sequence analysis, safety, and probiotic properties assessment. Whole genome sequencing results demonstrated that *L. plantarum* LPJZ-658 consists of 3.26 Mbp with a GC content of 44.83%. A total of 3254 putative ORFs were identified. Of note, a putative bile saline hydrolase (BSH) (identity 70.4%) was found in its genome. In addition, the secondary metabolites were analyzed, and one secondary metabolite gene cluster was predicted to consist of 51 genes, which verified its safety and probiotic properties at the genome level. Additionally, *L. plantarum* LPJZ-658 exhibited non-toxic and non-hemolytic activity and was susceptible to various tested antibiotics, indicating that *L. plantarum* LPJZ-658 was safe for consumption. Moreover, the probiotic properties tests confirm that *L. plantarum* LPJZ-658 also exhibits tolerance to acid and bile salts, preferably hydrophobicity and auto-aggregation, and excellent antimicrobial activity against both Gram-positive and Gram-negative gastrointestinal pathogens. In conclusion, this study confirmed the safety and probiotic properties of *L. plantarum* LPJZ-658, suggesting it can be used as a potential probiotic candidate for human and animal applications.

## 1. Introduction

Probiotics have a long history of use. They have gained global consensus for their health-regulating effects, driving the rapid development of the probiotic industry. *L. plantarum* has one of the largest genomes known among the lactic acid bacteria (LAB) and is a facultative heterofermentative LAB [1]. It may obtain energy from different sugars, and it reflects high adaptability to a variety of niches such as dairy products, vegetables, wine, and the gastrointestinal of humans and animals [2]. Among them, *L. plantarum* has good physiological properties and probiotic functions and has become a hot spot for research. A large number of studies have confirmed that *L. plantarum* has probiotic effects such as antioxidant, dyslipidemia regulation [3], intestinal inflammation and barrier function improvement [4,5], and intestinal homeostasis regulation [6]. In addition, *L. plantarum* strains are capable of producing various antimicrobial compounds, such as organic acids, hydrogen peroxide, and antiaflatoxigenic [7,8], to against a wide range of pathogenic bacteria [9,10].

On the other hand, the excellent properties of *L. plantarum*, such as its tolerance to acidic pH [11], resistance to the gastrointestinal tract [12], and adhesion to the intestinal mucosa [13], have also made *L. plantarum* can have beneficial effects on the host health. Furthermore, the connection between humans and *L. plantarum* has been further strengthened by altering the brain parameters and host immunity. Thus, *L. plantarum* can be used to prevent or treat certain allergic [14], depressive [15], and Alzheimer’s disease [16], as a potential therapy for neurological and psychological disorders [17], with great potential for application development.

Although few adverse events or safety issues has been reported, the potential safety issues of probiotic still attract concern. In 2001, an expert consultation convened under the auspices of the World Health Organization (WHO) and the Food and Agriculture Organization (FAO) proposed a beneficial definition of probiotics. Later in 2014, the definition was refined to “live microorganisms that, when administered in adequate amounts, confer a health benefit on the host” and further stated that all probiotics must be “safe for their intended use [18,19,20]”. Furthermore, probiotics can survive and proliferate in the gastrointestinal tract and have a strong internal transfer capacity [21], so the development of new probiotic strains must be evaluated for safety before use. In this study, the novel *L. plantarum* LPJZ-658 isolated and screened from naturally fermented dairy products was evaluated, and the safety of the *L. plantarum* LPJZ-658 was evaluated by genome sequencing, hemolysis test, drug resistance test, and acute oral toxicity test. In addition, the probiotic properties of the *L. plantarum* LPJZ-658 were evaluated by tolerance test to acid and bile salt conditions, antibacterial test, hydrophobicity, and auto-aggregation test, aiming to provide a theoretical basis for the development of new functional probiotic products.

## 2. Materials and Methods

### 2.1. Bacterial Strain

*L. plantarum* LPJZ-658 was isolated from naturally fermented dairy products. The genome has been sequenced and submitted to the NCBI database (NCBI no. SRR22306760). *Lactobacillus rhamnosus* GG (LGG) was purchased from American Type Culture Collection (ATCC, 53103, St. Cloud, MN, USA). *Salmonella typhimurium* (*S. typhimurium*, BNCC333565), *Escherichia coli* (*E. coli*, BNCC269342), and *Staphylococcus aureus subsp. aureus Rosenbach* (*S. aureus*, BNCC310011) was purchased from BeNa Culture Collection, Beijing, China. *L. plantarum* LPJZ-658 and LGG were cultured in de Man-Rogosa-Sharpe (MRS) broth (Beijing Solarbio Science & Technology Co., Ltd., Beijing, China). S. *typhimurium* and *E. coli* were cultured in Lysogeny broth (LB) broth (Sangon Biotech Shanghai Co., Ltd., Shanghai, China). *S. aureus* was cultured in Brian Heart Infusion (BHI) broth (Qingdao Hi-Tech Industrial Park Hope Bio-Technology Co., Ltd., Qingdao, China). All the cultures were grown in broth overnight at 37 °C before use.

### 2.2. Genome Sequencing and Bioinformatics Analysis of L. plantarum LPJZ-658

The genomic DNA of *L. plantarum* LPJZ-658 was extracted with the SDS method [22], and the whole genome of *L. plantarum* LPJZ-658 was sequenced using Illumina NovaSeq PE150 at the Beijing Novogene Bioinformatics Technology Co., Ltd. (Beijing, China) and assembled with SOAPdenovo and SPAdes and ABySS software. Coding gene were predicted using GeneMarkS software (V4.17) (http://topaz.gatech.edu/GeneMark/) (accessed on 25 March 2020) [23]; repetitive sequences were predicted using RepeatMasker (Version open-4.0.5) software and TRF (Tandem Repeats Finder, V4.07b) [24,25]; transfer RNA (tRNA) prediction by tRNAscan-SE software (V1.3.1) [26], ribosomal RNA (rRNA) prediction by RNAmmer software (V1.2) [27], and small RNA (sRNA) determination by the program “cmsearch” (V1.1rc4) [28,29].

The Gene function was predicted based on seven databases, including Non-Redundant Protein Database (NR) [30], Clusters of Orthologous Groups (COG) [31], Kyoto Encyclopedia of Genes and Genomes (KEGG) [32,33], Gene Ontology (GO) [34], Transporter Classification Database (TCDB) [35], and Swiss-Prot [36]. The SignalP [37] database was used to predict the secretory proteins, while the EffectiveT3 [38] software was used to predict the Type I-VII proteins secreted by the pathogenic bacteria. Meanwhile, the secondary metabolism gene clusters were predicted by the antiSMASH [39]. Additionally, the pathogenicity and drug resistance analyses were performed by Antibiotic Resistance Genes Database (ARDB) [40], Virulence Factors of Pathogenic Bacteria (VFDB) [41], and Pathogen Host Interactions (PHI) [42]. Active enzymes involved in carbohydrate metabolism were predicted by Carbohydrate-Active Enzymes (CAZy) Database [43].

### 2.3. Evolutionary Analysis

The Basic Local Alignment Search Tool (BLAST) of the GenBank (NCBI) was used to determine similarity values. To indicate species identification, sequences with ≥97% similarity to the previously published sequences were utilized as the criterion. Based on the 16S rRNA gene sequence analysis, a phylogenetic tree was constructed. The analysis included 20 nucleotide sequences, including 1 sequence of *Lactobacillus* strains obtained in this study and 19 sequences belonging to *Lactobacillus* species obtained from the GenBank. The Neighbour-Joining method was used to generate evolutionary history in Molecular Evolutionary Genetics Analysis (MEGA) 11 software. Bootstrapping was confided for 1000 replicates, and only bootstrap values above 50% were shown.

### 2.4. Safety Test of L. plantarum LPJZ-658

#### 2.4.1. Hemolytic Activity of *L. plantarum* LPJZ-658

The hemolytic activity of *L. plantarum* LPJZ-658 was detected as previously described with some modification [44]. *L. plantarum* LPJZ-658 was streaked onto a Columbia blood agar plate (Beijing Land Bridge Technology Co., Ltd., Beijing, China) and incubator at 37 °C for 48 h, the hemolysis of the colony of *L. plantarum* LPJZ-658 was observed based on the zone of hemolysis around the colonies. The β-hemolytic *S. aureus* and the γ-hemolytic LGG [45] were used as control strains.

#### 2.4.2. Antibiotic Susceptibility of *L. plantarum* LPJZ-658

The antibiotic susceptibility of *L. plantarum* LPJZ-658 was detected using a disk diffusion method recommended by the Clinical and Laboratory Standard Institute (CLSI) [46]. Briefly, the overnight grown culture of *L. plantarum* LPJZ-658 was evenly coated on the MRS broth and incubated for 20 h at 37 °C. A total of 100 μL of the bacterial solution was evenly streaked onto MRS agar plates until dry. Then, antibiotic discs (Hangzhou Binhe Microorganism Reagent Co., Ltd., Hangzhou, China) were attached to the medium surface with sterile forceps, and the plates were incubated upside down in a 37 °C incubator for 24 h. After incubation, the diameter of the inhibition zones was measured. LGG and *S. aureus* were used as control.

#### 2.4.3. Oral Toxicity Analysis of *L. plantarum* LPJZ-658

Eight- to ten-week-old C57BL/6 mice were purchased from Jilin GENET-MED Biotechnology Co., Ltd. (Jilin, China). The oral toxicity of *L. plantarum* LPJZ-658 was analyzed as previously described [47]. Briefly, 20 male and 20 female C57BL/6 mice were adaptive feeding for 5 days before the experiment, and then male and female mice were divided into two groups, with 10 males and 10 females in each group, respectively. *L. plantarum* LPJZ-658 was given by oral gavage once a day at a dose of 10^11^ CFU/day for 14 days. The behavior, death, and poisoning of the mice were observed, and the average weight per mouse was calculated once a day.

### 2.5. In Vitro Characterization of L. plantarum LPJZ-658

#### 2.5.1. Acid and Bile Salt Tolerance Analysis of *L. plantarum* LPJZ-658

The acid and bile salt tolerance of *L. plantarum* LPJZ-658 was detected as previously described with some modification [48]. For acid tolerance of *L. plantarum* LPJZ-658, the active grown cells were harvested, washed with phosphate-buffered saline (PBS) buffer, and resuspended in an equal volume of MRS broth with pH adjusted to 3.0, and then 100 µL aliquots were obtained at different time intervals (0 h and 3 h) for gradient dilution. Suitable dilutions were selected for inoculation on MRS agar and incubated for 24 h at 37 °C. Colonies were counted and recorded.

Bile tolerance of *L. plantarum* LPJZ-658 was examined in MRS broth containing 0.3% (*w*/*v*) bile salts (Beijing Solarbio Science & Technology Co., Ltd., Beijing, China). Samples were collected at 0 h and 3 h, respectively. Bacteria were serially diluted 10-fold using PBS and inoculated in triplicate onto MRS agar plates. The plates were incubated at 37 °C for 24 h. Colonies were counted and recorded. The above two experiments were repeated three times each, respectively.

#### 2.5.2. Cell Surface Hydrophobicity Analysis of *L. plantarum* LPJZ-658

Cell surface hydrophobicity of *L. plantarum* LPJZ-658 was evaluated as the reported method with some modifications [49]. The bacteria were grown in MRS broth at 37 °C for 20 h and then harvested by centrifugation at 1500× *g* for 20 min. The pellets were washed and resuspended in PBS buffer (pH 7.4) to the optical density (OD_600_) corresponding to about 10^8^ CFU/mL (*A*_0_). Then, an equal volume of xylene was added and incubated at 37 °C for 10 min, and the suspensions were vortexed for 2 min. The two-phase system was incubated at 37 °C for 1 h. The aqueous phase was measured by determining the OD at 600 nm (*A*_1_). The percentage of bacterial adhesion to solvent was calculated: [(*A*_1_ − *A*_0_)/*A*_0_] × 100.

#### 2.5.3. Auto-Aggregation Analysis of *L. plantarum* LPJZ-658

Auto-aggregation analysis of *L. plantarum* LPJZ-658 was performed, referring to the reported method with some modifications [50]. Bacteria cultures were grown at 37 °C in MRS broth for 20 h. Bacteria were then harvested by centrifugation, washed two times, and resuspended in PBS buffer. The initial concentration of *L. plantarum* LPJZ-658 was adjusted to OD_600_ to obtain a viable bacterial count (*A*_0_) of approximately 10^8^ CFU/mL. Absorbance (*At*) was then measured using the same method after 24 h of incubation at 37 °C. The auto-aggregation percentage was expressed as: [(*A*_t_ − *A_0_*)/*A*_0_] × 100.

#### 2.5.4. Antibacterial Activity

The inhibitory effects of *L. plantarum* LPJZ-658 against common bacterial pathogens, such as *E. coli* BNCC269342, *S. typhimurium* BNCC333565, and *S. aureus* BNCC310011 were measured using the Oxford cup diffusion method [51] with LGG strain as control. *E. coli, S. typhimurium*, and *S. aureus* were incubated in LB medium at 37 °C for 18 h. Bacterial pathogens suspension was diluted to 10^8^ cfu/mL, and 0.2 mL of each diluted pathogens suspension was inoculated on LB plates and evenly with a sterile cotton swab, creating a vertical Oxford cup on each equidistant plate. Then, 200 μL of triple concentrated *L. plantarum* LPJZ-658 or LGG culture supernatant was added to the hole, respectively. Distilled water was chosen as the control. Each strain had three replicates. The plates were then incubated for 24 h at 37 °C. The antimicrobial function of *L. plantarum* LPJZ-658 was determined by measuring the diameter of the circular antimicrobial zone.

### 2.6. Statistical Analysis

Paired *t*-tests were performed on all data using GraphPad Prism 7. Data are expressed as means ± standard error of the mean (SEM), and *p*-values of <0.05 were considered significant.

## 3. Results

### 3.1. Genome Characteristics of L. plantarum LPJZ-658

The genome sequencing (Table 1) indicated that the whole genome of *L. plantarum* LPJZ-658 comprised 3.26 Mbp (3,259,902 bp), with an average GC content of 44.83%, encoding a total of 3254 ORFs. The total length of all encoded genes was 2,733,840 bp, and the average length of encoded genes was 840 bp, containing 100 Tandem Repeat (TR) sequences with a total length of 20,760 bp, including 78 Minisatellite DNAs with a total length of 9131 bp, and 1 Microsatellite DNA with a total length of 39 bp, 69 tRNA genes, 10 5S rRNAs, 5 16S rRNAs and 23S rRNAs, and 1 sRNA. The genome sequencing data for L. plantarum LPJZ-658 has been submitted to GenBank (no. SRR22306760), and the *L. plantarum* LPJZ-658 genome map is shown in Figure 1A.

### 3.2. Genomic Functional Annotation of L. plantarum LPJZ-658

The genomic functional annotation of *L. plantarum* LPJZ-658 was predicted based on GO, KEGG, COG, NR, CAZy, T3SS, VFDB, CARD, PHI, ARDB, Swiss-Prot and Pfam databases (blastp, evalue ≤ 1 × 10^−5^, identity ≥40%, and coverage ≥40%). The final annotation results are shown in Table 2. A total of 4161 genes of the *L. plantarum* LPJZ-658 genome were annotated. Among them, more genes were functionally annotated in NR, Swiss-Prot, KEGG, COG, GO, and Pfam databases, with 3189, 1255, 3106, 2359, 2211, and 2211 genes, respectively and accounting for 98%, 38.56%, 95.45%, 72.49%, 67.94%, and 67.94% of the total genes, respectively. Notably, only one annotated gene was available in the ARDB database, accounting for 0.03% of the total number of genes.

The functional annotation to COG showed that 22 categories were classified. The most numerous clarified COG categories were transcription (244 genes, accounting for 10.34% of the total number of annotated genes), followed by carbohydrate transport and metabolism (240 genes, accounting for 10.17% of the total number of annotated genes), general function prediction (237 genes, accounting for 10.04% of the total number of annotated genes), amino acid transport and metabolism (231 genes, accounting for 9.79% of the total number of annotated genes) (Figure 1B). The results of KEGG annotation showed that a total of 1318 genes corresponding to the KEGG pathway were enriched in 34 metabolic pathways (Figure 1C). Among them, carbohydrate metabolism, membrane transport, and amino acid metabolism were the three most important metabolic pathways. A total of 132 genes encode protein structural domains belonging to the CAZy family of carbohydrates (Figure 1D). Among them, 62 were Glycoside Hydrolases (GH), 40 were Glycosyltransferases (GT), 23 were Carbohydrate- Binding Modules (CBM), 6 were Carbohydrate Esterases (CE), and 1 was Auxiliary Activities (AA).

### 3.3. Phylogenetic Analysis and Identification of L. plantarum LPJZ-658

The phylogenetic relationships between *L. plantarum* LPJZ-658 and 19 *Lactobacillus* strains obtained from GenBank are described as shown in the phylogenetic tree based on 16S rRNA gene sequence analysis. *L. plantarum* is grouped and distinguishable from other strains of Lactobacillus (Figure 1E). The strain *L. plantarum* LPJZ-658 isolated in this study clustered together and was monophyletic with *L. plantarum* JCM 1149, with a bootstrap value of 100%.

### 3.4. Safety Evaluation of L. plantarum LPJZ-658

#### 3.4.1. Acute Oral Toxicity Study of *L. plantarum* LPJZ-658 (14 Days Repeated Dose)

All mice were divided into two groups according to gender, and challenged with 10^11^ CFU/mice of *L. plantarum* LPJZ-658 for 14 consecutive days. As seen in Table 3, no significant differences were found in the average body weight of mice challenged with *L. plantarum* LPJZ-658 (*p* < 0.05). Furthermore, no death or gross pathological findings were observed in mice of either group.

#### 3.4.2. Virulence-Related Genes and Hemolytic Activity of *L. plantarum* LPJZ-658

Safety evaluation is an important step in screening new probiotic candidates for human and animal applications, although the results of the virulence factors analysis of *L. plantarum* LPJZ-658 predicted a total of 99 virulence-related factors. However, almost all of these genes showed low similarity, with less than 75% identity (Appendix A). The phenotypic analysis further confirmed the hemolytic activity of *L. plantarum* LPJZ-658 (Figure 2). After 48 h of incubation on Columbia Blood Agar Plat, neither *L. plantarum* LPJZ-658 nor LGG exhibited hemolysis ability, whereas *S. aureus* showed significant β-hemolytic activity. All these results indicated *L. plantarum* LPJZ-658 is non-hemolysis, which is a prerequisite characteristic of probiotics.

#### 3.4.3. Antibiotic Resistance

The antibiotic susceptibility of *L. plantarum* LPJZ-658 is shown in Table 4. The results demonstrated that *L. plantarum* LPJZ-658 is sensitive to ampicillin, erythromycin, gentamicin, midecamycin, and streptomycin; moderately resistant to chloramphenicol and clindamycin; and resistant to ciprofloxacin, kanamycin, and norfloxacin. The Antibiotic Resistance Genes analysis of *L. plantarum* LPJZ-658 was performed using CARD antibiotic resistance gene databases with identity >75%. Only one antibiotic-resistance gene was identified (Appendix A).

### 3.5. Functionality Tests as Potential Probiotics of L. plantarum LPJZ-658

#### 3.5.1. Tolerance of *L. plantarum* LPJZ-658 to Acidic Conditions and Bile Salts

The effect of the simulated harsh gastrointestinal environment on the potential of *L. plantarum* LPJZ-658 in comparison to LGG is shown in Table 5. The survival of *L*. *plantarum* LPJZ-658 was found to be ≥90% after 3 h exposure to acidic conditions (pH 3.0) and 0.3% bile salt solution. Additionally, *L. plantarum* LPJZ-658 exhibited no significant differences compared to LGG in both acidic conditions and bile salt solution. Genomic features explaining the observed bile tolerance were identified in the bile saline hydrolase (BSH) genome of *L. plantarum* LPJZ-658 (70.4% identity).

#### 3.5.2. Cell Surface Hydrophobicity of *L. plantarum* LPJZ-658

Hydrophobicity and auto-aggregation are important indicators for probiotics adherence to the intestine. As seen in Table 6, after 24 h of incubation, the hydrophobicity capacity of *L. plantarum* LPJZ-658 and LGG was 33.50% and 28.04%, respectively. Additionally, the auto-aggregation capacity of *L. plantarum* LPJZ-658 and LGG was 62.59% and 52.32%, respectively. Interestingly, the percentages of hydrophobicity and auto-aggregation of *L. plantarum* LPJZ-658 were significantly higher than those of LGG.

#### 3.5.3. Antibacterial Activity

One of the important characteristics of probiotics is to antagonize pathogens. In the current study, the antibacterial activity of *L. plantarum* LPJZ-658 was evaluated against Gram-positive (*S. aureus*) and Gram-negative (*S. typhimurium* and *E. coli*) pathogenic bacteria. The *L. plantarum* LPJZ-658 exhibited excellent antagonistic activity against the tested bacteria, in which the inhibition spectrum of *E. coli* and *S. typhimurium* were significantly better than that of LGG (Table 7), and among these, *L. plantarum* LPJZ-658 showed the strongest antibacterial ability. In order to determine if *L. plantarum* LPJZ-658 is capable of producing secondary metabolites, the online tool antiSMASH was used to screen the *L. plantarum* LPJZ-658 genome for secondary metabolite biosynthetic gene clusters. The results showed that a secondary metabolite gene cluster consisting of 51 genes was predicted in the genome of *L. plantarum* LPJZ-658. All 51 genes encode proteins belonging to type III PKS (T3PKS) (Figure 3). All these results indicated *L. plantarum* LPJZ-658 own good probiotic properties.

## 4. Discussion

In this study, a new probiotic named *L. plantarum* LPJZ-658 has been isolated and characterized. The application of whole-genome sequencing-based safety analysis is critical to the development of the probiotic industry. Probiotic bacteria belonging to *Lactiplantibacillus plantarum* generally have several properties that can be predicted by the presence of the corresponding genes. These are, in particular, genes involved in the synthesis of bacteriocins [52], metabolism [53], conjugation of unsaturated fatty acids [54], production of biologically active peptides [55], and genes encoding exopolysaccharides [56]. In addition, genome-based safety analysis may help to identify potential risk factors for candidate probiotics. However, it should be knowledge that environmental conditions are closely related to gene expression. Therefore, the safety analysis of genomes only theoretically reveals the risk level of probiotics.

To verify the safety of *L. plantarum* LPJZ-658, the hemolytic activity was tested, and it was reported that probiotics should not have hemolytic properties, especially β-hemolysis, which is considered harmful [57]. Furthermore, it is recommended to assess the hemolytic activity of probiotics used in food products, even if they are generally recognized as safe (GRAS) [58,59]. In our study, *L. plantarum* LPJZ-658 exhibits γ-hemolysis. Additionally, in the process of safety evaluation of probiotics, resistance assessment is one of the important criteria for strain screening. The resistance of probiotics is mainly intrinsic and acquired. The intrinsic resistance developed during the formation of microorganisms is not usually transferred. Contrarily, if the strain has acquired resistance, it has the risk of transferring virulence genes into the intestine [60]. Therefore, it is necessary to study the distribution of antibiotic-resistance genes in probiotics and the antibiotic resistance pattern of strains [61]. Similar to commercial probiotic strain LGG, the same resistance gene was identified for *L. plantarum* LPJZ-658 in the CARD database with coverage >40% and identity >75% [48]. Furthermore, the antibiotic susceptibility of *L. plantarum* LPJZ-658 was further investigated, and the results indicated that *L. plantarum* LPJZ-658 was sensitive or intermediate sensitive to seven antibiotics, including ampicillin, erythromycin, gentamicin, midecamycin, streptomycin, chloramphenicol, and clindamycin. In vitro, assessment of virulence traits is a prerequisite for probiotic strain candidates, and in vivo studies in the appropriate animal models are also essential to validate the safety of probiotic strain candidates. Oral toxicity studies were considered to be the standard for testing the safety of bacterial strains [62]. In this study, mice orally gavaged with high concentrations of *L. plantarum* LPJZ-658 for 14 days showed good health status, and no acute toxicity was detected, which confirms that *L. plantarum* LPJZ-658 is safe for in vivo application. Additionally, more clinical trials are needed in the future to comprehensively evaluate the safety of *L. plantarum* LPJZ-658.

In order to bring out the health benefits of LPJZ-658, probiotic potential has been further investigated. Tolerance to low-acidic gastric and bile-rich intestinal conditions is one of the prerequisites for probiotic candidate strains due to the extreme conditions that provide a stressful environment for bacteria [63]. The tolerance of probiotic candidate strain to gastrointestinal conditions is normally measured at 3 h after co-culture, as food is transported along the human intestine for up to 3 h [64]. Additionally, bile salts were known to inhibit the growth and multiplication of bacteria, so tolerance to bile salts allows probiotic strains to survive, grow, and function during gastrointestinal transport [65]. In this study, *L. plantarum* LPJZ-658 maintained high levels of survival in a simulated digestive fluid (pH 3.0 and 0.3% bile salt), which is similar to commercial probiotic LGG. As known, BSH is an enzyme that breaks down bile salts [66]. In addition, BSH is an important enzyme for cholesterol removal, which is associated with cholesterol scavenging ability and is beneficial for people on a high-fat diet [67]. Therefore, bile tolerance in *L. plantarum* LPJZ-658 may be ascribed to the presence of BSH (identity 70.4%) encoding genes, which could theoretically determine the ability of the strain to tolerate intestinal conditions [68]. It is worth noting that in our previous studies, *L. plantarum* LPJZ-658 supplementation protects against WD/CCl4-induced non-alcoholic steatohepatitis progression, which is accompanied by alteration of bile acid metabolism profiles. This may be related to the higher BSH activity of LPJZ-658. The ability of probiotics to adhere to the intestine is associated with several types of interactions, including hydrophobicity and auto-aggregation [69]. A prerequisite for being an ideal candidate probiotic is its ability to adhere to epithelial cells and mucosal surfaces. Xylene was chosen as the polar solvent in this study because it reflects the hydrophobicity and hydrophilicity of the cell surface [70], and *L. plantarum* LPJZ-658 exhibited higher hydrophobicity towards xylene than LGG, indicating good bacterial adhesion of *L. plantarum* LPJZ-658 to hydrocarbons. These properties are essential for the colonization of probiotic cultures in the gastrointestinal (GI) epithelium to prevent elimination by peristalsis and to play a functional role in intestinal homeostasis [64]. Auto-aggregation is also closely related to cell adhesion to the GI tract, which explains the probiotic properties of the bacteria [71]. The present study revealed that *L. plantarum* LPJZ-658 had higher cell hydrophobicity and auto-aggregation compared to LGG, indicating its good cell adhesion properties. *S. thyphimurium*, *E. coli*, and *S. aureus* are common intestinal pathogens that are highly susceptible to causing infections in the body, promoting an inflammatory response in the intestinal tract, destroying the intestinal barrier, and causing diarrhea and other diseases [72]. Previous studies had reported the antimicrobial activity of probiotics [73], which in agreement with our study, the *L. plantarum* LPJZ-658 showed excellent antimicrobial activity against pathogenic bacteria, including *S. thyphimurium*, *E. coli*, and *S. aureus*. The production of antimicrobial substances is one of the important factors in regulating intestinal microecology and maintaining the health of the host. The antimicrobial substances in probiotics include organic acids, hydrogen peroxide, and bacteriocins [74,75]. The intriguing property of *L. plantarum* LPJZ-658 is the presence of a secondary metabolite gene cluster consisting of 51 genes found in its genome. All of these 51 genes encode proteins belonging to T3PKS. T3PKS is one of the two most abundant biosynthetic gene clusters in all LAB genera [76]. As known, T3PKS are all known to be small dimeric proteins (80–90 kDa) associated with the biosynthesis of polyketides [77]. Polyketides are natural metabolites that comprise the basic chemical structure of antibiotics, antifungals, parasiticides, and immunomodulators [78]. This class of compounds mainly includes polyethers, tetracycles, quinones, macrolides, and other substances with great applications in anti-infective, antitumor, and immunosuppressive applications [79]. It has also been suggested that the specific secondary metabolite T3PKS may be related to bacterial survival and antibacterial activity in the intestinal environment [80,81]. This suggests that *L. plantarum* LPJZ-658 encoded T3PKS may be correlating with its broad-spectrum antibacterial activity.

## 5. Conclusions

In this study, the genomic level, safety, and probiotic properties of *L. plantarum* LPJZ-658 were comprehensively evaluated using in vitro and in vivo methods. All these results indicated that *L. plantarum* LPJZ-658 has great potential as a probiotic in human and animal applications. This was supported by genome sequencing and other normal probiotic characteristics such as non-hemolytic activity, high survival under acid and bile conditions, high antibiotic susceptibility, good cell adhesion ability, and excellent antimicrobial activity. In addition, no pathogenicity or mortality was observed in oral toxicity studies. Overall, *L. plantarum* LPJZ-658 is safe and has good probiotic properties, and more studies could be done in the future to clarify the application potential of *L. plantarum* LPJZ-658 on gastrointestinal health and in the food industry.

## Figures and Tables

**Figure 1 microorganisms-11-01620-f001:**
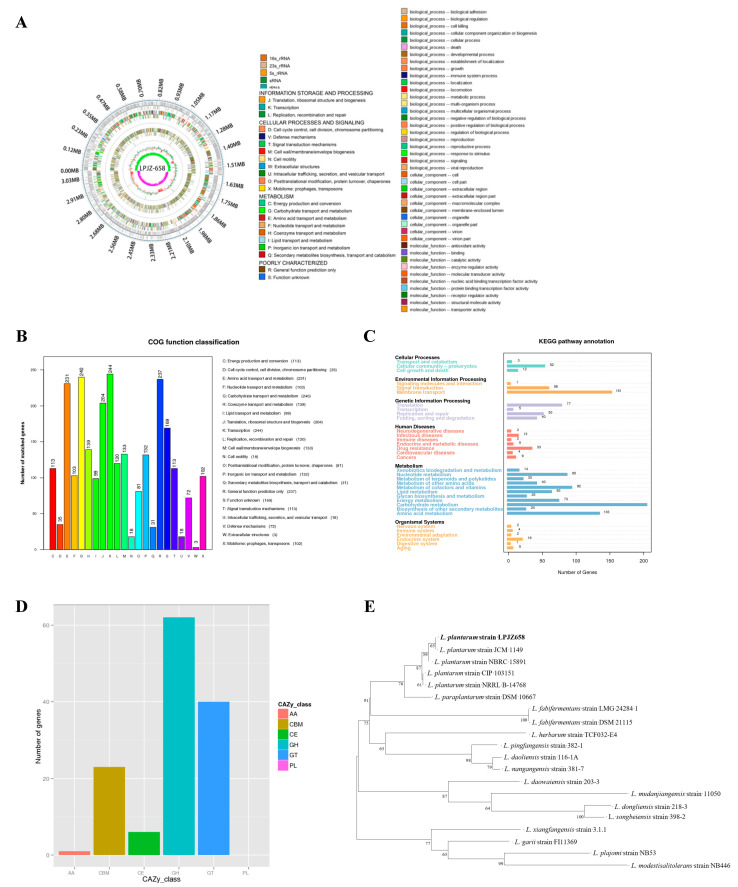
Genome features of *L. plantarum* LPJZ-658. (**A**) Genomic map of *L. plantarum* LPJZ-658. From the outer circle to the inner circle: the first circle is the distribution of the coding genes. The second is the COG annotated genes. The third is the KEGG annotated genes. The fourth is the GO annotated genes. The fifth circle is the distribution of ncRNA. (**B**) The amino acid sequence of *L. plantarum* LPJZ-658 annotation based on the COG database. The left side shows the distribution of the number of COG functions analyzed on the annotations. The horizontal coordinates are the COG function categories. Different characters and colors correspond to different types on the right side, and the vertical coordinates are the number of genes. (**C**) The amino acid sequence of *L. plantarum* LPJZ-658 annotation based on the KEGG database. The number on the bar chart indicates the number of genes on the annotation. The other coordinate axis is the code for each level 1 functional class in the database. (**D**) The amino acid sequence of *L. plantarum* LPJZ-658 annotation based on the CAZy database. (**E**) The phylogenetic relationship among selected Lactobacillus strains and *L. plantarum* LPJZ658 was generated by using the neighbor-joining method based on the 16S rRNA gene sequence.

**Figure 2 microorganisms-11-01620-f002:**
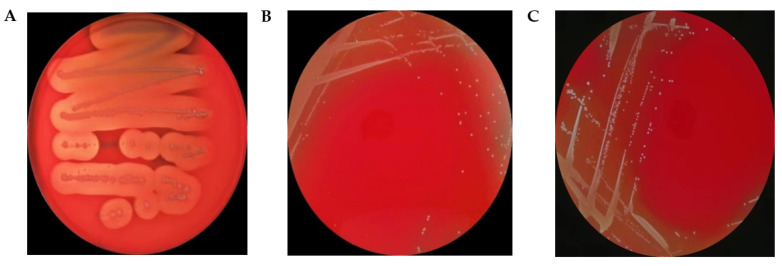
Hemolytic activity of *L. plantarum* LPJZ-658. As a positive control, *S. aureus* (**A**) produced an obvious zone of β-haemolysis *L. plantarum* LPJZ-658 (**B**), and LGG (**C**) showed γ-hemolysis.

**Figure 3 microorganisms-11-01620-f003:**
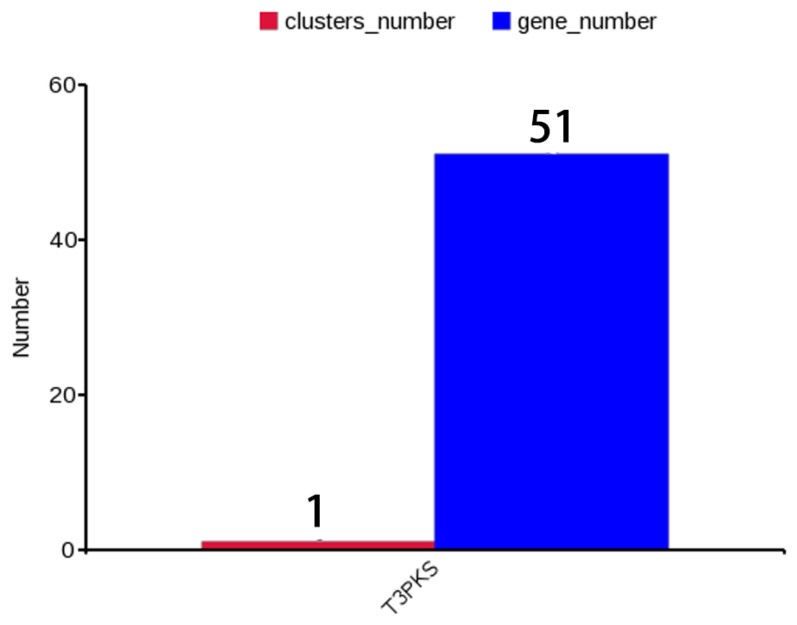
A statistical plot of secondary metabolic gene clusters and the number of corresponding genes of *L. plantarum* LPJZ-658.

**Table 1 microorganisms-11-01620-t001:** Main genome characteristic of *L. plantarum* LPJZ-658.

Characteristic	Value
Genome size (bp)	3,259,902
GC content (%)	44.83
Number of genes	3254
Total length of all genes (bp)	2,733,840
Average gene length (bp)	840
Number of tRNA	69
Number of rRNA	16
Number of sRNA	1

**Table 2 microorganisms-11-01620-t002:** Database distribution of functional gene annotation from the *L. plantarum* LPJZ-658.

Type	Gene Number
NR	3189
GO	2211
KEGG	3106
COG	2359
CAZy	122
T3SS	161
VFDB	99
PHI	156
ARDB	1
Swiss-Prot	1255
Pfam	2211
Secretory_Protein	45
TCDB	258
CARD	113

**Table 3 microorganisms-11-01620-t003:** Relative dynamic change of mice challenged with *L. plantarum* LPJZ-658 for 14 days.

Sex	Number	Number of Deaths	Poisoning Symptoms	Anatomic Abnormality	Mortality	Body Weight (g)		
						0 d	7 d	14 d
Male	10	0	None	None	0	22.40 ± 0.25	23.32 ± 0.41	23.88 ± 0.50
Female	10	0	None	None	0	19.82 ± 0.33	20.94 ± 0.29	21.57 ± 0.43

**Table 4 microorganisms-11-01620-t004:** Antibiotic susceptibility of *L. plantarum* LPJZ-658 to antibiotics.

Types	Antibiotics	Susceptibility	
		LPJZ-658	*S. aureus*
β-lactam antibiotics	Ampicillin	S	S
Aminoglycosides	Gentamicin	S	S
	Kanamycin	R	S
	Streptomycin	S	S
Macrolides	Erythromycin	S	R
	Midecamycin	S	R
Quinolones	Ciprofloxacin	R	S
	Norfloxacin	R	S
Amphenicols	Chloramphenicol	I	I
Lincosamides	Clindamycin	I	R

Antibiotics: Ampicillin (10 μg); Gentamicin (10 ± 2.5 μg); Kanamycin (30 μg); Streptomycin (10 μg); Erythromycin (15 μg); Midecamycin (30 μg); Ciprofloxacin (5 μg); Norfloxacin (10 μg); Chloramphenicol (30 μg); Clindamycin (2 μg). S: sensitive; I: intermediate sensitive; R: resistant.

**Table 5 microorganisms-11-01620-t005:** The survival of *L. plantarum* LPJZ-658 in simulated gastrointestinal conditions.

Bacterial Strains	Survival Rate (%)	
	Acid Tolerance (pH 3.0)	Bile Tolerance (0.3%)
LGG	93.58 ± 4.28%	88.66 ± 2.13%
LPJZ-658	95.71 ± 2.70%	90.52 ± 2.93%

**Table 6 microorganisms-11-01620-t006:** Cell surface hydrophobicity of *L. plantarum* LPJZ-658.

Bacterial Strains	Hydrophobicity (%)	Auto-Aggregation (%)
LGG	28.04 ± 1.30% ^b^	52.32 ± 1.07% ^b^
LPJZ-658	33.50 ± 1.23% ^a^	62.59 ± 1.29% ^a^

^a,b^: Different superscript letters in the same column indicate statistical differences between LGG and *L. plantarum* LPJZ-658 at the level of *p* < 0.05. Values are represented as mean ± SEM.

**Table 7 microorganisms-11-01620-t007:** Antibacterial Activity of *L. plantarum* LPJZ-658.

Bacterial Strains	*S. aureus*	*E. coli*	*S. typhimurium*
LGG	12.000 ± 0.707	11.700 ± 0.447 ^b^	13.667 ± 0.983 ^b^
LPJZ-658	14.917 ± 2.178	14.000 ± 0.632 ^a^	16.833 ± 1.125 ^a^

^a,b^: Different letters on the shoulder of data in the same column indicate statistical differences between LGG and *L. plantarum* LPJZ-658 at the level of *p* < 0.05. Values are represented as mean ± SEM.

## Data Availability

Data sharing is not applicable to this article.

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
