# Peer review of "Genome Sequence and Evaluation of Safety and Probiotic Potential of Lactiplantibacillus plantarum LPJZ-658"

_microorganisms, 2023, doi:10.3390/microorganisms11061620_

Round 1

Reviewer 1 Report

The research topic is relevant. The study was generally well-planned. The results have a certain scientific value. Nevertheless, there are a number of remarks. Probiotic bacteria belonging to Lactiplantibacillus plantarum generally have a number of properties that can be predicted by the presence of the corresponding genes. These are, in particular, genes involved in the synthesis of bacteriocins, metabolism and conjugation of unsaturated fatty acids, production of biologically active peptides (for example, dipeptides), genes encoding exopolysaccharides. Although these genes were not detected in the studied strain, it is desirable to discuss this in terms of alternative probiotic properties. The secondary metabolite gene cluster was predicted by the authors. Thus, it also requires more discussion regarding the expected probiotic effects associated with secondary metabolites. Please explain the possible mechanisms of the antibacterial activity of the strain. Since there is a patent ("Application of Lactobacillus plantarum LPJZ-658 in preparation of medicines for treating non-alcoholic fatty liver disease. CN113786419A"), please discuss potential mechanisms for strain effects on the liver based on predicted genes.

Please add literary references to the latest publications of 2023 on the topic. It is highly recommended to use the correct strain names: Lactiplantibacillus plantarum LPJZ-658 and Salmonella Typhimurium. Also, specify whether BSH or bile acids/salts are metabolized ("the ability of L. plantarum LPJZ-658 to metabolize BSH") and correct the figure number (incorrect "Figure 5") to "Figure 3".

The manuscript may be recommended for publication after a minor revision.

Author Response

We would like to express our gratitude for the opportunity to revise our manuscript entitled “Genome sequence and evaluation of safety and probiotic potential of Lactobacillus plantarum LPJZ-658” (Manuscript ID: microorganisms-2423342). We sincerely thank all the reviewers for their valuable feedback, which we have carefully considered and used to improve the quality of our work.

We have addressed the reviewers’ comments in a point-by-point manner below, and we have made the necessary revisions to the manuscript. Specific concerns raised by the reviewers have been numbered for clarity. Our responses are given in normal font, and changes/additions to the manuscript are highlighted in red.

Thank you again for your time and consideration.

Response to the comments of Reviewer 1

Point 1: The research topic is relevant. The study was generally well-planned. The results have a certain scientific value. Nevertheless, there are a number of remarks. Probiotic bacteria belonging to Lactiplantibacillus plantarum generally have a number of properties that can be predicted by the presence of the corresponding genes. These are, in particular, genes involved in the synthesis of bacteriocins, metabolism and conjugation of unsaturated fatty acids, production of biologically active peptides (for example, dipeptides), genes encoding exopolysaccharides. Although these genes were not detected in the studied strain, it is desirable to discuss this in terms of alternative probiotic properties. The secondary metabolite gene cluster was predicted by the authors. Thus, it also requires more discussion regarding the expected probiotic effects associated with secondary metabolites. Please explain the possible mechanisms of the antibacterial activity of the strain. Since there is a patent ("Application of Lactobacillus plantarum LPJZ-658 in preparation of medicines for treating non-alcoholic fatty liver disease. CN113786419A"), please discuss potential mechanisms for strain effects on the liver based on predicted genes.

Response 1: We sincerely appreciate the valuable comments. We have followed the relevant discussion according to the Reviewer’s suggestion.

Point 2: Please add literary references to the latest publications of 2023 on the topic. 

Response 2: We sincerely appreciate the valuable comments. We have cited more recent references into the revised manuscript.

Point 3: It is highly recommended to use the correct strain names: Lactiplantibacillus plantarum LPJZ-658 and Salmonella Typhimurium.

Response 3: We feel great thanks for your professional review work on our article. We have used the correct strain names “Lactiplantibacillus plantarum LPJZ-658” and “Salmonella Typhimurium” in the revised manuscript.

Point 4: Also, specify whether BSH or bile acids/salts are metabolized ("the ability of L. plantarum LPJZ-658 to metabolize BSH") . 

Response 4: We feel sorry for our inaccurate discussion. We have changed it to “bile tolerance in L. plantarum LPJZ-658 may be ascribed to the presence of BSH (identity 70.4%) encoding genes”.

Point 5: Correct the figure number (incorrect "Figure 5") to "Figure 3".

Response 5: We feel sorry for our carelessness. We have corrected this mistake in the revised manuscript.

Best Wishes to you!

Yours sincerely,

Cuiqing Zhao

Reviewer 2 Report

This is an apparently interesting study since it investigates when considering the importance of the discovery of new probiotic species and its characterization all with the importance of future application for human and animal. Article is well written, it treats an actual problem, it is readable and the concept is easy to follow. The scope of work is defined properly. Modern methodology was applied and the results are presented with clarity and detail. I suggest acceptance of the manuscript in its present form.

Author Response

We would like to express our gratitude for the opportunity to revise our manuscript entitled “Genome sequence and evaluation of safety and probiotic potential of Lactobacillus plantarum LPJZ-658” (Manuscript ID: microorganisms-2423342). We sincerely thank all the reviewers for their valuable feedback, which we have carefully considered and used to improve the quality of our work.

We have addressed the reviewers’ comments in a point-by-point manner below, and we have made the necessary revisions to the manuscript. Specific concerns raised by the reviewers have been numbered for clarity. Our responses are given in normal font, and changes/additions to the manuscript are highlighted in red.

Thank you again for your time and consideration.

Response to the comments of Reviewer 2

This is an apparently interesting study since it investigates when considering the importance of the discovery of new probiotic species and its characterization all with the importance of future application for human and animal. Article is well written, it treats an actual problem, it is readable and the concept is easy to follow. The scope of work is defined properly. Modern methodology was applied and the results are presented with clarity and detail. I suggest acceptance of the manuscript in its present form.

Response: Thank you for your comments on our manuscript.

Best Wishes to you!

Yours sincerely,

Cuiqing Zhao
